# Learning Eye-in-Hand Camera Calibration
# from a Single Image

**Eugene Valassakis**[*]
The Robot Learning Lab
Imperial College London
United Kingdom
pev115@ic.ac.uk

**Kamil Drezckowski**[*]
The Robot Learning Lab
Imperial College London
United Kingdom
krd115@ic.ac.uk

**Edward Johns**
The Robot Learning Lab
Imperial College London
United Kingdom
e.johns@imperial.ac.uk

**Abstract:** Eye-in-hand camera calibration is a fundamental and long-studied problem in robotics. We present a study on using learning-based methods for solving this problem online from a single RGB image, whilst training our models with entirely synthetic data. We study three main approaches: one direct regression model that directly predicts the extrinsic matrix from an image, one sparse correspondence model that regresses 2D keypoints and then uses PnP, and one dense correspondence model that uses regressed depth and segmentation maps to enable ICP pose estimation. In our experiments, we benchmark these methods against each other and against well-established classical methods, to find the surprising result that direct regression outperforms other approaches, and we perform noise-sensitivity analysis to gain further insights into these results.

**Keywords:** Camera Calibration, Robot Manipulation, Sim-to-Real

## 1 Introduction

Eye-in-hand setups, where images from a camera rigidly mounted to the end-effector are used to control a robot, are a popular choice for precise robot manipulation. Compared to control using a fixed external camera, the eye-in-hand setup enables precise control because images can be captured very close to an object, as well as generalisation across the workspace when actions are defined relative to the camera or end-effector [1, 2, 3]. To apply to precise manipulation tasks, these methods require ac-

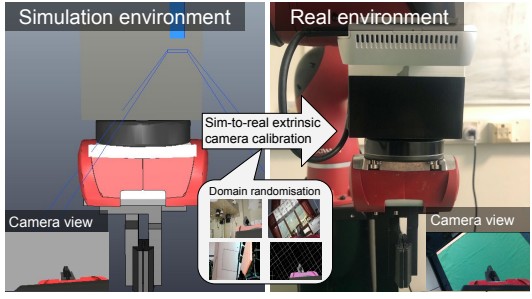

Figure 1: Illustration of our task and setup.

curate eye-in-hand extrinsic calibration, to ensure that actions or poses inferred in the camera's frame can be transformed to the robot's frame. Whilst calibration-free approaches exist, such as end-to-end control [4, 5], these are only suitable for model-free policy learning and cannot be used for methods that involve explicit planning or reasoning about 3D space.

Classically, solving the eye-in-hand camera calibration problem is performed off-line [6, 7, 8, 9] and involves the use of a specific external calibration object, such as a checkerboard or an AprilTag. Poses of the calibration object can be estimated by the camera, and by recording them from several robot configurations, the rigid-body transformation between the camera and end-effector can be estimated analytically. Although with perfect information these methods can yield exact calibrations, they are in practice limited by the quality of the data they use, including how precisely the pose of the calibration object can be estimated or the accuracy of the recorded end-effector poses. Furthermore, these methods are unsuited for use by non-experts since they require dedicated external calibration objects and a new calibration dataset to be carefully collected for each re-calibration. Finally, they are impractical as re-calibration needs to be performed every time the camera position changes due to (1) re-mounting, (2) wear and tear, and (3) collisions between the camera and the environment. Deep learning methods can alleviate these drawbacks. Given the initial overhead required to generate training data and to train a model, a deep learning-based method can be readily

---

[*]Joint First Author Contribution

5th Conference on Robot Learning (CoRL 2021), London, UK.

used to re-calibrate a camera online from a single image using parts of the robot as the calibration object. This potential has been demonstrated for *eye-to-hand* camera calibration, where an external camera observes the robot from a distance [10, 11].

To the best of our knowledge, there are currently no studies on using deep learning methods for *eye-in-hand* calibration, which is the focus of this paper. We propose that this calibration can be estimated directly from the camera's image itself, without requiring any external apparatus, as long as the image can observe part of the end-effector, such as the gripper's fingers (see fig. 1). Based on this, we identify and benchmark three natural approaches for leveraging the power of deep learning in this way: two based on classical, geometric pose estimation where deep learning is used in place of components that are typically manually engineered, and one which uses deep learning to directly regress the camera calibration matrix. We show in our real-world experiments that the two approaches that rely on geometric analysis perform poorly, and we analyse empirically the structural limitations and challenges that cause this. We also show that, perhaps surprisingly, direct regression outperforms all alternatives, including classical methods based on fiducial markers with automatic data collection. Finally, given that direct regression can estimate the calibration matrix from a single image online, we propose and evaluate a method for fusing multiple estimates of the camera's pose to increase the accuracy of the aggregated estimate.

As such, our work has three key contributions: (1) we investigate the suitability of three natural alternatives for using deep-learning for enabling eye-in-hand camera calibration from a single image, (2) we show the potential of end-to-end deep learning for online eye-in-hand camera calibration in everyday environments, and (3) we provide an analysis for the two learning-based methods that rely on geometric approaches, through which we discover why these methods may not be suitable for eye-in-hand camera calibration. An accompanying video can be found at
https://www.robot-learning.uk/learning-eye-in-hand-calibration .

## 2   Related Work

Eye-in-hand camera calibration is a long-studied problem, with significant advances introduced in the 1990's, with what are now well-established solutions. The problem consists of inferring the camera to end-effector pose for a camera mounted on the wrist of a robotic manipulator. To do so, several end-effector to robot base poses are recorded, along with corresponding estimates of calibration object to camera poses. The problem is then reduced to solving the $AX = XB$ equation through a formalism introduced by Shiu and Ahmad in [12]. Several works then followed that mainly differ in their strategy for solving the $AX = XB$ equation. For instance, Tsai and Lenz [7] improve on the efficiency of Shiu and Ahmad's solution by proposing a closed-form solution. Park and Martin [8] and Dornaika and Harod [9] also consider such a closed-form solution while relying on Lie theory and unit quaternions respectively.

While the above approaches solve for rotation and translation separately, Dornaika and Harod [9] also propose a non-linear technique for solving for both simultaneously. Daniilidis [6] proposes a solution to the problem using dual quaternions. Finally, more recent works study various extensions for particular settings such as using structure for motion for calibration [13], using model-based pose estimation and tracking for online calibration [14], simultaneously considering the data time synchronisation problem [15], calibrating depth sensors with non-overlapping views [16], automatically doing both calibration pattern localisation and eye-in-hand camera calibration [17], and optimising the robot kinematics parameters and the camera to end-effector pose simultaneously [18].

Recently, deep learning methods have had great successes on closely related tasks such as pose estimation [19, 20] and end-to-end robotics manipulation from wrist-mounted cameras [1, 4]. Moreover, sim-to-real transfer has shown promise in alleviating the large data requirements that hinder the scalability of such methods [4, 21, 22, 23]. In this work, we investigate the naturally emerging question of how one might use such methods for eye-in-hand camera calibration, which is fundamental to a large number of robotics pipelines. Closest to our work, Lee et al. [11] and Labbe et al. [10] investigate two different deep learning approaches for eye-to-hand calibration of a camera looking straight at a robot. [11] is based on keypoint regression followed by Perspective-n-Point (PnP) [24] while [10] adopts a render-and-compare approach. In our work, we consider the eye-in-hand setup that is more common when precise manipulation needs to be achieved by looking closely at the end-effector of the robot [4, 25]. This problem also differs in its properties from the eye-to-hand setup,

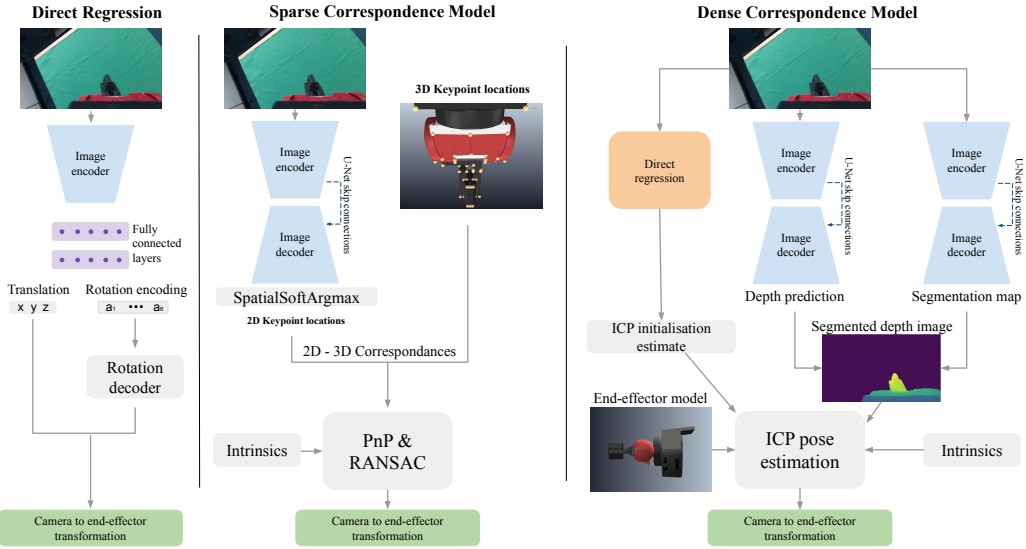

Figure 2: From left to right: Illustration of our Direct Regression, Sparse Correspondence and Dense Correspondence models.

since in a typical image there is now a much more constrained view that only allows for the tip of the end-effector to be visible, and used as an anchor for pose estimation.

# 3 Methods

In this section, we present three methods that are conceptually promising for eye-in-hand camera calibration from a single RGB image, and which also raised interesting and novel scientific questions that we discuss in sections 4 and 5. The first method is an end-to-end direct regression method that estimates the camera's pose from an RGB image. In contrast, the other two methods use deep learning to provide key missing components for well-established geometric pose estimation approaches. The latter have been shown to work well in other problem settings [11, 26, 27, 28], but eye-in-hand camera calibration has the particularity that not only the camera is too close to the end-effector for typical vision sensors to capture a depth image of it, but also the visible geometry varies widely within our inference space. To test whether it is still feasible to use such approaches, our second method uses deep learning to regress the 2D locations of predefined keypoints on the end-effector, and passes them to the PnP [24] algorithm to solve for the camera's pose. The main challenge with this method is that the large variations in the visible geometry lead to keypoint occlusions and to some keypoints being often outside of the image frame. This is illustrated in fig. 3. Our final method tests whether we can alleviate this issue by using deep learning to regress a segmented depth image of the end-effector and an initial guess of the camera's pose, and the Iterative Closest Point (ICP) [29, 30, 31] algorithm to refine this initial estimate. All three methods are illustrated in fig. 2.

## 3.1 Problem Setting

Our problem setting consists of the typical eye-in-hand camera calibration problem, where the aim is to estimate the camera's extrinsic matrix, which is the camera to end-effector pose, $T_{EC} = [R_{EC}|t_{TC}] \in SE(3)$, where $R_{EC} \in SO(3)$ and $t_{EC} \in \mathbb{R}^3$ is the orientation and position of the camera in the end-effector frame respectively. We also define the end-effector to robot base pose as $T_{BE}$, the calibration object to camera pose as $T_{CO}$, and an image captured by the wrist-mounted camera as $I$. Throughout this paper, we use $\tilde{\cdot}$ to denote an estimated quantity.

As opposed to classical approaches, we constrain ourselves to eye-in-hand camera calibration from a single RGB image without any external apparatus, enabling our methods to be readily deployed in the wild. We further constrain ourselves to using only synthetic data in order to alleviate the costly data requirements of deep learning approaches, and to obtain ground truth labels that facilitate effective learning. Finally, we assume that we have an accurate estimate of the camera intrinsic

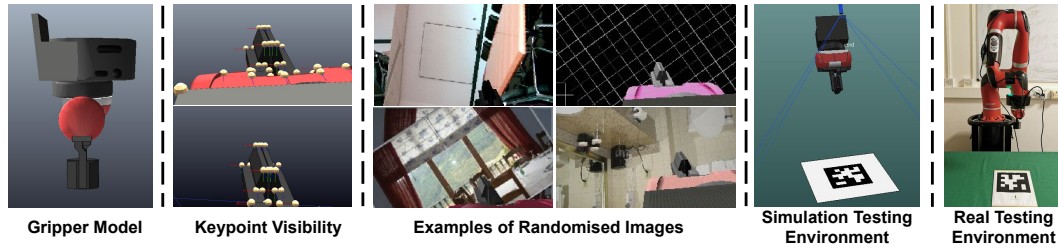

| Gripper Model | Keypoint Visibility | Examples of Randomised Images | Simulation Testing Environment | Real Testing Environment |

Figure 3: Left to right: Model of the gripper. Illustration of keypoint visibility from different viewpoints. Examples of randomised simulation images. Simulation and real world testing environments.

matrix, which the manufacturer typically provides, and access to the CAD model of our robot's end-effector.

## 3.2 Direct Regression Model

Direct regression is the end-to-end deep learning method we consider. As illustrated in the left diagram of fig. 2, a neural network (NN) is tasked with regressing the camera's pose from a single RGB image and is trained in simulation using ground truth labels in a supervised setting. The only architectural constraint we introduce comes from the parameterisation of the orientation, where we use the 6D rotation encoding introduced in [32].

Overall, our network outputs a 9 dimensional tensor, with 3 dimensions representing position and 6 the orientation. It consists of a convolutional encoder followed by linear layers that regress the camera pose. We trained it with the ground truth labels using the Mean Squared Error (MSE) loss, and we fully breakdown the architecture and training details in our supplementary material.

## 3.3 Sparse Correspondence Model

Our sparse correspondence model is illustrated in the middle of fig. 2. Starting from an RGB image, it uses a U-Net [33] type architecture with a spatial-soft-argmax [34] output activation in order to regress a set of 2D keypoints in image space. We use the sum of an L1 and MSE losses to train our network on the ground truth labels, such that each output dimension predicts the 2D location of a predefined 3D keypoint. A detailed breakdown of our architecture and training parameters can be found in our supplementary material.

At test time, we use the network's output, the corresponding 3D keypoint locations defined in the object's model, and the camera's intrinsic matrix to estimate the camera's pose with PnP and RANSAC [24], which are a popular pose estimation and outlier rejection algorithms.

Finally, in order to alleviate the issue of some keypoints being out of frame depending on the viewpoint, we (1) consider 38 keypoints that are present in at least 70% of the training images, and (2) for any keypoint that remains out of frame, we train the model to predict its projection in the image plane. This way, at test time, we can infer which keypoints are out of frame by looking at the predictions at the rim, and ignoring these predictions for PnP pose estimation.

## 3.4 Dense Correspondence Model

Our Dense Correspondence model is illustrated on the right of fig. 2. Starting from an RGB image, it uses three independently trained NNs to estimate the camera's pose in two stages. In the first stage, the three NNs are used to predict (1) a depth map, (2) a segmentation mask of the gripper, and (3) an ICP initialisation. In the second stage, (1) the segmentation mask is used to segment the depth map to only return depth values on gripper pixels, (2) the camera intrinsic matrix is used to project the segmented depth map to a point cloud, and (3) ICP is used to establish dense correspondences between the estimated point cloud and the model of the end-effector and to refine the camera's pose estimate from the first stage. The network used for ICP initialisation is the same as the one used for direct regression. The same U-Net architecture as for the sparse correspondence model is used to predict the depth and segmentation maps. See our supplementary material for further architecture design choices and training details.

### 3.5 Fusing Multiple Estimates

Compared to classical eye-in-hand camera calibration methods, which require calibration to be done offline and with special apparatus, the proposed deep learning alternatives can estimate the camera's pose online in unstructured environments. This creates the possibility of fusing multiple estimates to increase the overall calibration accuracy. In order to explore this, we implemented a simple aggregation algorithm that rejects 20% of the least likely samples from a set of candidate estimates under a Gaussian data model. It then averages the remaining samples together to yield the final estimate. Our algorithm for this procedure is available in our supplementary material.

### 3.6 Dataset Generation and Sim-to-Real Transfer

We train our networks entirely with simulated data using the Coppelia [35] simulator. In order to generate the dataset, we randomise the pose of the camera relative to the end-effector at each simulated timestep and record (1) the current RGB, depth and end-effector segmentation images, (2) the ground truth extrinsic matrix, and (3) the 2D keypoint locations in image space.

In order to overcome the "reality gap", we apply visual domain randomisation [21, 22, 23]. Specifically, we (1) randomise the colours/textures of the simulated gripper and the light sources on the simulator, (2) replace the background of our images with random images of textures and indoor scenes [36, 37], and (3) apply a post-processing colour jitter operation [38], further randomly perturbing the brightness, contrast, saturation and hue of the whole image. In total, we generated 10 000 labelled images in approximately 30 minutes, examples of which can be seen in fig. 3.

## 4 Experiments

In our experiments, we evaluate how the different proposed deep learning approaches compare to each other and to various standard off-the-shelf calibration methods both in simulation and in the real world. From them emerges the surprising result that simple direct regression using end-to-end deep learning outperforms both the classical approaches tested and the correspondence-based deep learning methods. As such, we perform a series of analysis experiments, which we describe in section 5, that are aimed at gaining insights into this surprising observation.

In our comparisons, we include all our deep learning-based methods and the following established methods readily available on OpenvCV [39]: Dainiilidis et al. [6], Tsai and Lenz [7], Doraika and Harod [9], and Park and Martin [8]. For all experiments, we use the Realsense D435 (or its simulated counterpart) at a $480 \times 848$ resolution, and for the deep learning methods we downsample these images to a resolution of $144 \times 256$.

Comparing classical methods to our deep learning ones fairly can be somewhat challenging since classical methods require a training set of several end-effector to robot and calibration object to camera poses to perform a single calibration, while our methods only require a single image. In sections 4.1 and 4.2 we describe our procedures and what we did to medicate this issue, and in our supplementary material we provide our algorithm for each of these procedures.

### 4.1 Simulation Experiment

Our simulated environment consists of a wrist-mounted camera attached to a Sawyer robot's gripper that is free to move around an AprilTag (see fig. 3). To benchmark all methods, we (1) collect a dataset $\{I_i, T^i_{BE}, \tilde{T}^i_{CO}\}^{15}_{i=1}$ of images and corresponding AprilTag and end-effector poses, with the tag poses $\tilde{T}_{CO}$ estimated using the AprilTags3 library [40, 41, 42], (2) use all of the 15 datapoints in this dataset to estimate the extrinsic matrix using each classical method, and evaluate these estimates, (3) use each of the 15 datapoints in this dataset independently to estimate the extrinsic matrix using each of the proposed learned methods, and evaluate each of the single image estimates, and (4) for each of the learned methods, we fuse together all 15 independent predictions using our fusion procedure described in 3.5 and evaluate the fused estimate.

We repeat this procedure for 100 different ground truth extrinsic matrices, take the average and standard deviation, and display the results in table 1. We evaluate translation and rotation independently: Given an estimate of the extrinsics $\tilde{T}_{EC} = [\tilde{R}_{EC}|\tilde{t}_{EC}]$, with ground truth $T^*_{EC} = [R^*_{EC}|t^*_{EC}]$, we

|  | Simulation | | Real World |
| Method | $\epsilon_t$ [mm] | $\epsilon_R$ [degrees] | $\epsilon_{std}$ [mm] |
|---|---|---|---|
| TSAI [7] | $(171.4 \pm 271.2)$ | $(15.9 \pm 20.6)$ | $(17.5 \pm 18.9)$ |
| PARK [8] | $(68.7 \pm 70.1)$ | $(6.3 \pm 4.7)$ | $(15.3 \pm 14.3)$ |
| HORAUD [9] | $(68.9 \pm 72.2)$ | $(6.3 \pm 4.7)$ | $(16.3 \pm 16.6)$ |
| DANIILIDIS [6] | $(110.8 \pm 122.8)$ | $(11.5 \pm 29.4)$ | $(14.7 \pm 13.6)$ |
| DR | $\mathbf{(13.4 \pm 4.1)}$ | $\mathbf{(4.4 \pm 1.4)}$ | $(10.6 \pm 4.1)$ |
| DR (fusion) | $\mathbf{(13.4 \pm 4.1)}$ | $\mathbf{(4.4 \pm 1.4)}$ | $\mathbf{(10.4 \pm 4.0)}$ |
| SC | $(363.6 \pm 353.1)$ | $(117.4 \pm 48.2)$ | $(516.4 \pm 338.1)$ |
| SC (fusion) | $(338.5 \pm 301.0)$ | $(120.2 \pm 42.7)$ | $(291.2 \pm 129.6)$ |
| DC | $(93.5 \pm 45.9)$ | $(33.9 \pm 25.7)$ | $(21.0 \pm 10.4)$ |
| DC (fusion) | $(73.4 \pm 21.2)$ | $(23.1 \pm 13.9)$ | $(15.0 \pm 1.7)$ |

Table 1: Evaluation of the classical methods, our Direct Regression (DR), Sparse Correspondence (SC) and Dense Correspondence (DC) methods, and their fusion variants with aggregated estimates.

define the position error as $e_t = ||\tilde{t}_{EC} - t^*_{EC}||_2$ , where $|| \cdot ||_2$ is the L2 norm, and the rotational error as $e_R = \theta$, the angle from the axis-angle representation of the rotation matrix $R_\Delta = (w, \theta)$ that satisfies the relationship $R^*_{EC} = R_\Delta \tilde{R}_{EC}$, where $w$ and $\theta$ are the axis and angle of rotation.

## 4.2 Real World Experiment

The real-world evaluation environment is analogous to the simulated environment and consists of a Sawyer robot with a wrist-mounted camera moving around an AprilTag (see fig. 3). In order to evaluate our methods in the real world, we (1) collect a training data bank $\mathcal{D}_{train} = \{I_i, T^i_{BE}, \tilde{T}^i_{CO}\}_{i=1}^{40}$ and an evaluation dataset $\mathcal{D}_{eval} = \{T^i_{BE}, \tilde{T}^i_{CO}\}_{i=1}^{60}$ automatically by scripting a trajectory around the AprilTag, where AprilTag poses are estimated using the AprilTags3 library [40, 41, 42], (2) we sample 40 training datasets of 15 datapoints each from our training data bank, (3) we use all 15 datapoints in each training set to get an estimate of the camera extrinsic matrix using each classical method, and evaluate that estimate, (4) we use each one of the 15 datapoints in each dataset to get an estimate of the extrinsic matrix using our learned methods, and evaluate each single estimate, and (5) for each of the learned methods, we fuse together all 15 independent predictions using the procedure described in 3.5 and evaluate the fused estimate.

We repeat this procedure for two different ground truth extrinsics, calculate the average and standard deviation, and display the results in table 1. Since ground truth extrinsic parameters are not available in the real world, we take inspiration from [6, 7] and use an indirect error metric: For each corresponding end-effector to robot pose and calibration object to camera pose from the evaluation dataset, $\{T^i_{BE}, \tilde{T}^i_{CQ}\} \in \mathcal{D}_{eval}$, we estimate the pose of the calibration object in the robot's frame, $\tilde{T}^i_{BO} = T^i_{BE}\tilde{T}^i_{EC}\tilde{T}^i_{CO} = [\tilde{R}^i_{WO}|\tilde{T}^i_{BO}]$, where $\tilde{T}_{EC}$ is the estimate of the camera to end-effector pose that we are evaluating. We then define the error metric $\epsilon_{std} = \left(1/60 \sum_{i=1}^{60} ||\tilde{t}^i_{BO} - \mu||_2^2\right)^{1/2}$ as the standard deviation of the estimated calibration object position, where $\mu = 1/60 \sum_{i=1}^{60} \tilde{t}^i_{BO}$ is the estimated mean object position.

We emphasise that although the AprilTag is visible in all images in both the simulated and real environments, it is never used to help the calibration of our deep learning methods. We use these images simply to ensure that we have the exact same calibration and evaluation datasets for both classical and our learned methods.

## 4.3 Results

All benchmarking results are shown in table 1. Our main observation is that the direct regression method outperforms all others, which is a surprising result. First, it was not expected to outperform the classical methods since they use analytical solutions to estimate the extrinsics, which in the absence of noise in the system should give a perfect calibration. We believe this stems from using automatic data gathering for the classical methods, which does not allow for carefully curating the calibration dataset and the distribution of poses within it, a manual trial-and-error process that is

generally required to bolster calibration accuracy. This also raises the question about the sensitivity of these methods to said noise, which we investigate in section 5.

Second, it was also not expected to outperform the correspondence-based learned methods, since those introduce strong geometric constraints that intuitively should help guide the model to good solutions. We thoroughly investigate the reasons behind this observation in sections 5.2 and 5.3.

# 5 Analysis

In this section, we aim to build a deeper understanding of the surprising results that we observed in our experiments, which showed that a direct regression of calibration parameters outperforms methods that incorporate well-understood geometric modelling. We split this section into three parts, each analysing the performance of one particular approach.

## 5.1 Classical Methods

To better understand the results from our experiments in section 4, we first aim to assess how sensitive classical methods are to sources of noise in the system. Assuming the main source of error stems from the calibration object pose estimation, we fix every other source of noise in simulation to their ground truth values. We then vary the noise injected to the calibration object's poses by controlled amounts and observe how this affects the quality of the calibration result from classical methods.

Precisely, we vary the noise levels from $0mm$ to $10mm$ and $0°$ to $10°$, in increments of $0.5mm$ and $0.5°$, and for each noise tier, we perform calibration with each of the classical methods considered. We then compare the result to the ground truth extrinsics and obtain position errors. We repeat this procedure for 100 camera extrinsics, with the average errors obtained and their standard deviations illustrated in the top graph of fig. 4.

We can see that, as expected, with perfect information, the classical methods return an exact solution. However, we also see that the calibration quality rapidly deteriorates with increased noise in the calibration object's poses. This illustrates our motivation for using a deep learning-based approach, since our methods are independent of any test-time data gathering or calibration object pose estimation quality.

Figure 4: Top: Analysis of sensitivity to noise in the estimated tag positions of classical methods. Bottom: Analysis of sensitivity to noise in the estimated keypoint positions of PnP+RANSAC.

## 5.2 Sparse Correspondences

In order to better understand why our sparse correspondence model did not perform strongly we set up another controlled experiment. Starting from ground truth 2D-3D correspondences, we add fixed amounts of noise to the 2D keypoint locations and observe the effect of this on the quality of PnP + RANSAC pose estimation. Specifically, we iterate through 200 random extrinsic matrices, and for each {2D keypoints, 3D keypoints, extrinsic} tuple, we perform PnP + RANSAC with the noise-injected 2D keypoints and compare the result to the ground truth. We repeat this for different magnitudes of noise, ranging from 0 to 9 pixels in increments of 1 pixel, and averaged over all the datapoints considered. We plot the resulting errors in the "70% in frame" curve in the bottom graph of fig. 4.

During our investigation we also considered using a smaller number of keypoints, but ones that always remain in frame. With those we observed that even though this is an easier task for the network prediction, our final extrinsics estimation did not improve. In order to understand this behaviour, we repeat the controlled noise experiment but only considering the keypoints that always appear in frame in the images, which resulted in 12 remaining keypoints clustered on the gripper's fingers. This is plotted in the "100% in frame" curve in the bottom graph of fig. 4.

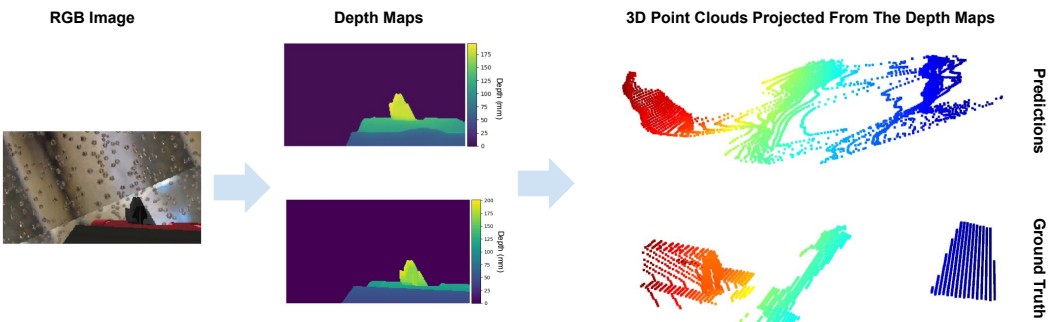

Figure 5: Left: RGB image of the gripper rendered in simulation. Top centre: Depth image predicted by our model from the simulated RGB image. Top right: point cloud projected from the predicted depth image. Bottom centre: Depth image rendered in simulation from the same viewpoint. Bottom right, point cloud projected from the ground truth depth image.

From fig. 4, we clearly see that there is a very high sensitivity of sparse correspondence pose estimation to errors in 2D keypoint pixel locations. For the keypoints in our training set, to get an extrinsic calibration position error of less than 1cm, our networks' average keypoint prediction error would need to be less than 4 to 5 pixels. This is with the additional challenge of the network having to keep consistent predictions even though sometimes the keypoints appear out of frame. On the other hand, if we consider the easier to learn problem of only predicting keypoints that always appear in-frame, the sensitivity skyrockets after single-pixel average error in 2D keypoint locations. Overall, we believe that this sensitivity to noise is the primary reason for why the sparse correspondence model did not give a strong performance.

## 5.3 Dense Correspondences

Although conceptually using a dense geometric correspondence approach should alleviate some of the drawbacks of using sparse correspondences, we found that in practice our model that uses ICP to refine estimates made by the direct regression method decreases their accuracy.

The reason for this becomes apparent when considering the illustrations shown in fig. 5. Although the depth values obtained by the neural networks look appealing when projected onto the image plane, they fall short when considering the full 3D structure. We conjecture that the main reason for this is that in order to recover a correct 3D structure from such close distances, depth values need to have strong discontinuous jumps that typical neural networks have difficulty modelling. This is supported by the observations in fig. 5: We can see that the overall depth increase seems to match the ground truth values, but there is an averaging effect that tends to make the depth values increase smoothly, which does not allow us to recover the correct shape. As such, we can conclude that using a simple depth regression technique in this setting is inappropriate for this task's requirements, with further research needed to overcome the discontinuity problem.

## 6 Conclusion

In this work, we presented and evaluated three deep learning-based methods for online eye-in-hand camera calibration from a single image. We trained all our models entirely on synthetic images, and evaluated them against each other and long-established calibration methods in simulation and the real world. Surprisingly, our experiments indicated that a direct regression method from images to camera extrinsics outperformed other alternatives. In order to better understand this result, we then conducted a series of introspection experiments, which indicated that a strong shortcoming of geometry-based methods seems to be their sensitivity to noise in their input data. Finally, while depth map regression for dense correspondences conceptually could have provided the answer to this, our experiments indicated that due to pronounced discontinuities in the depth profiles of such close-up viewpoints, further research is needed, specifically in learning depth prediction for eye-in-hand robot manipulation.

**Acknowledgments**

This work was supported by the Royal Academy of Engineering under the Research Fellowship scheme.

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
