# OpenReview forum: "Learning Eye-in-Hand Camera Calibration from a Single Image"
_robot-learning.org/CoRL/2021/Conference — CoRL2021 Poster_

### Official Review · Reviewer_NfZb · 2021-07-15

**Originality:** Good
**Technical Quality:** Very Good
**Clarity Of Presentation:** Very Good
**Impact:** 3

**Recommendation:**

Weak Accept: I recommend accepting the paper, but will not argue for my recommendation if the majority of other reviewers have a different opinion.

**Summary:**

The paper analyzes three methods of performing eye-in-hand calibration: 1) directly training a neural network to regress to the camera pose, 2) using a neural network to generate sparse keypoints and use Perspective-n-Points + RANSAC, and 3) using a neural network to generate dense correspondences and use ICP. Surprisingly, the authors found that direct regression, which achieves estimation error of 13.4mm and 4.4 degrees, significantly outperformed the other methods that have more informative geometric priors built in. The paper proceeds to analyze why this is the case and offers corresponding explanations – matching sparse keypoints seems too sensitive to noise, and predicted dense correspondences are unrealistically smooth, owing to neural networks’ interpolation bias.

**Issues:**

Please see strengths and weaknesses.

**Reviewer Expertise:**

Good: General knowledge of the area

**Strengths And Weaknesses:**

Strengths:
-	The paper conducts a well-focused study.
-	The paper offers surprising and practical insights.
-	The paper is clearly written.
-	Implementation and experiments seem well executed.

Weaknesses:
-	While the authors acknowledge that doing “excessive hyperparameter tuning” for the geometric methods may make the comparison unfair, I would still encourage the authors to try to tune their hyperparameters to see what their performance ceilings are.
-	Specifically, the authors may want to tune not just the network hyperparameters, but the ones for RANSAC and ICP.
-	The authors should discuss how much time it takes to generate the data and train the network.
- The authors should visualize more specific failure cases for each of the method to better illustrate their differences. This can be done in the supplementary materials.
-	Spacing and font of texts in Figure 2 could be improved.


**Summary Of Recommendation:**

My recommendation of weak accept is due to the clean and focused insight the paper provides, even though it does not introduce new algorithmic advances (e.g. proposing ways to mitigate the issues found in the correspondence-based approaches).

---

> ### Author Response · Authors · 2021-08-30
> **Answer to Reviewer NfZb**
>
> We appreciate the feedback from the reviewer and have answered the points raised below.  We have also now updated our paper and supplementary material, with changes reflected in blue.
>
> > While the authors acknowledge that doing “excessive hyperparameter tuning” for the geometric methods may make the comparison unfair, I would still encourage the authors to try to tune their hyperparameters to see what their performance ceilings are. Specifically, the authors may want to tune not just the network hyperparameters, but the ones for RANSAC and ICP
>
> We want to clarify that we have spent a significant amount of time tuning all of our methods to the best of our ability—this including tuning network architectures, training hyperparameters and the parameters of the RANSAC and ICP algorithms. To be specific, for RANSAC, we have tuned the number of iterations and the inlier threshold. For the ICP algorithm, we have tuned the number of ICP iterations, the error metric, the data association method, and the outlier rejection algorithm used. Overall, we believe that the most significant issues with these methods are as described in the Analysis section of our paper, rather than the need for additional hyperparameter tuning.
>
> > The authors should discuss how much time it takes to generate the data and train the network.
>
> Once the simulation environment and deep learning training code have been set up, generating the data for all models took approximately 30 minutes, while training a single deep learning model took up to a few hours depending on the method. We have now added a brief discussion about this in section 4, lines 92-96 of our supplementary material.
>
> > The authors should visualize more specific failure cases for each of the method to better illustrate their differences. This can be done in the supplementary materials
>
> We acknowledge that we do not showcase particular failure case examples in our paper and appreciate the constructive criticism. In practice, we found that they did not bring much insight into our discussion, and as such, we opted to discuss the underlying weakness of the different methods in the Analysis section of our paper instead.
>
>
> > Spacing and font of texts in Figure 2 could be improved.
>
> We appreciate the advice which we have accounted for in the updated version of Figure two in the revised paper.

---

> > ### Comment · Reviewer_NfZb · 2021-08-31
> > **Rebuttal Response**
> >
> > I would like to thank the authors for their clarifications in the response. My recommendation of weak accept remains unchanged.

---

### Official Review · Reviewer_SvQJ · 2021-07-23

**Originality:** Good
**Technical Quality:** Good
**Clarity Of Presentation:** Very Good
**Impact:** 3

**Recommendation:**

Weak Accept: I recommend accepting the paper, but will not argue for my recommendation if the majority of other reviewers have a different opinion.

**Summary:**

Eye-in-hand calibration estimates the pose of a camera rigidly mounted to some part of a robot, w.r.t. robot base. This paper explores 3 learning-based approaches for eye-in-hand calibration from a single RGB image. All training is done with synthetic images. Experiments compare the proposed approaches with each other and with traditional external marker-based optimization approaches, both in simulation and sim2real.

**Issues:**

- Sim-only experiments with some other gripper designs would make this paper stronger, but it is not necessary just for acceptance, in my opinion
- I do not expect the sparse correspondence pipeline suggestion to be followed up in this paper. That can be explored in another paper

**Reviewer Expertise:**

Very good: Comprehensive knowledge of the area

**Strengths And Weaknesses:**

# Strengths
- paper is well written and addresses a well-motivated problem
- experiments are thorough, and metrics are clearly described
- Figure 2 is particularly helpful to understand the paper

# Clarification
- Does the sparse correspondence model also output the identity of keypoints in addition to their location? If now, how are 2D-3D correspondences decided?

# Weaknesses
- It makes intuitive sense that the direct regression model works best among the 3 proposals in the regime of small viewpoint changes. The sparse correspondence system could have been better designed by taking inspiration from parallel literature in keypoint matching. For example, researchers are having much more success with detecting and matching keypoints between a pair of images, rather than directly regressing keypoints. The problem can be cast as detecting and matching keypoints from query image to an image taken from a canonical known pose with good visibility. Many references can be found in Section 2 of "COTR: Correspondence Transformer for Matching Across Images" (that particular paper is quite new, so I don't expect the authors of this paper to have used it, I am just providing it for is Related Works section.)
- A simulation-only analysis of the performance of these methods on different grippers would have verified whether the conclusions drawn here apply to all grippers, or just the Sawyer gripper.

**Summary Of Recommendation:**

I am in favour of accepting this paper because it introduces and thoroughly examines the idea of learning eye-in-hand calibration. However, I am a little skeptical of the conclusion regarding direct regression being significantly better than sparse correspondence **when** the sparse correspondence pipeline is more carefully designed (see above for a suggestion on how to do that).

---

> ### Author Response · Authors · 2021-08-30
> **Answer to Reviewer SvQJ**
>
> We thank the reviewer for their insightful comments regarding our paper. We have now uploaded a revised version of our paper and supplementary material with changes appearing in blue. Please find specific answers to the points raised below:
>
> > Does the sparse correspondence model also output the identity of keypoints in addition to their location? If now, how are 2D-3D correspondences decided?
>
> During training, the labels are generated such that each entry always corresponds to a specific keypoint. As such, from the network predictions is possible to associate each channel of the network's output with a specific 3D keypoint. We have now clarified this point in line 151 of the updated version of our paper.
>
> > It makes intuitive sense that the direct regression model works best among the 3 proposals in the regime of small viewpoint changes. The sparse correspondence system could have been better designed by taking inspiration from parallel literature in keypoint matching. For example, researchers are having much more success with detecting and matching keypoints between a pair of images, rather than directly regressing keypoints. The problem can be cast as detecting and matching keypoints from query image to an image taken from a canonical known pose with good visibility. Many references can be found in Section 2 of "COTR: Correspondence Transformer for Matching Across Images" (that particular paper is quite new, so I don't expect the authors of this paper to have used it, I am just providing it for is Related Works section.)
>
> We thank the reviewer for their insight and appreciate the suggestion. Indeed, it seems that casting the problem as a correspondence matching problem has the potential to yield better results, as well as naturally solving the issue of missing keypoints from the image plane. It will also be interesting to investigate how such a method might respond to a sim-to-real setting where the query image is taken from a different domain (simulation) than the target image (real world). As such, we value this as a great avenue for future work.
>
>
> > A simulation-only analysis of the performance of these methods on different grippers would have verified whether the conclusions drawn here apply to all grippers, or just the Sawyer gripper.  [...] Sim-only experiments with some other gripper designs would make this paper stronger, but it is not necessary just for acceptance, in my opinion
>
> We thank the reviewer for their constructive suggestion. We have now included this experiment in section 4.3, lines 155-166, of our supplementary material, where we test the performance of our best performing method on three additional grippers in simulation. These new grippers consist of a two-finger RG2 gripper, a three-finger Jaco Hand gripper and a Baxter suction gripper. In all three, our results are consistent with the results reported in our paper, which indicates there is nothing particular about the gripper tested in the paper and that the method would work consistently well on different hardware.

---

> > ### Comment · Reviewer_SvQJ · 2021-09-03
> > **post-rebuttal**
> >
> > Thank you for your response. I will keep my weak accept rating.

---

### Official Review · Reviewer_VdxP · 2021-07-25

**Originality:** Good
**Technical Quality:** Fair
**Clarity Of Presentation:** Good
**Impact:** 2

**Recommendation:**

Weak Accept: I recommend accepting the paper, but will not argue for my recommendation if the majority of other reviewers have a different opinion.

**Summary:**

The paper evaluates different methods for eye-in-hand calibration for the special case that a major part of the gripper is visible. Three alternative deep learning approaches are trained on synthetic data to solve this problem from a single RGB image: (1) Direct regression of the camera to end-effector pose (2) Depth prediction and ICP refinement with the gripper model (3) 2D Keypoint Detection + PnP + RANSAC. Also some classical baselines that detect a single April Tag from different robot poses are compared. In their experiments the direct regression method merged over some frames wins and the reason is found to be noise in the pose predictions of the gripper or marker.

**Issues:**

- Figure 4 top doesn't show green/blue lines
l325 "sythetic"

**Reviewer Expertise:**

Good: General knowledge of the area

**Strengths And Weaknesses:**

Strengths:

- The problem of single-image eye-in-hand calibration without external markers has not been tackled much, yet (maybe because the assumption that the gripper is always visible is too strong?)
- The approach compares several standard learning approaches that estimate the pose of the gripper in the camera frame and surprisingly shows that some are more accurate than external marker detections
- The simulation experiments on how the different calibration methods react to different types of noise are interesting
- The problem that current depth-sensors are often unable to sense at very close proximity is identified and therefore RGB is used
- It allows online calibration

Weaknesses:

- The resolution of the RGB sensor is not mentioned even though it has quite some impact on calibration results
- A single April Tag is not really adequate to get a precise calibration. In my experience calibration boards, e.g. the Charuco board lead to higher accuracies than reported in the paper with classical methods
- There is only one robotic experiment using a single camera-robot combination. More evidence is needed that this method consistently works well
- The paper mentions the Render & Compare approach from Labbe et al. but even though it is SoTA for hand-eye calibration, it is not tried here (I expect this to work better than directly regressing the pose).
- The assumption that the gripper is always in view is quite strong. In this case we could also just attach a marker to the gripper.
- For the constrained and un-occluded pose space of the gripper I would also expect classical edge matching techniques to work well
- The approach of pushing keypoints that are out of the image to the image border is questionable
- Creating the synthetic training data might be more tedious than doing the external calibration
- The achieved standard deviation of >1cm does not seem good enough to me

**Summary Of Recommendation:**

While the tackled problem could be interesting, the proposed evaluation is insufficient to draw any conclusions. The baselines and proposed approaches seem a bit too weak. I would recommend the authors to extend and resubmit their study to more setups, different markers and the render&compare approach or maybe some classical edge matching. For me the approach is also too dependent to have a sufficient part of the gripper in the image but in some specific cases it might make sense.

---

> ### Author Response · Authors · 2021-08-30
> **Answer to Reviewer VdxP: Part 4/4**
>
> References
>
> [1] Puang, E.Y., Tee, K.P. and Jing, W., 2020. KOVIS: Keypoint-based Visual Servoing with Zero-Shot Sim-to-Real Transfer for Robotics Manipulation. In 2020 IEEE/RSJ International Conference on Intelligent Robots and Systems (IROS) (pp. 7527-7533). IEEE.
>
> [2] Young, S., Gandhi, D., Tulsiani, S., Gupta, A., Abbeel, P. and Pinto, L., 2020. Visual imitation made easy. arXiv preprint arXiv:2008.04899.
>
> [3] Song, S., Zeng, A., Lee, J. and Funkhouser, T., 2020. Grasping in the wild: Learning 6dof closed-loop grasping from low-cost demonstrations. IEEE Robotics and Automation Letters, 5(3), pp.4978-4985.
>
> [4] Triyonoputro, J.C., Wan, W. and Harada, K., 2019, November. Quickly inserting pegs into uncertain holes using multi-view images and deep network trained on synthetic data. In 2019 IEEE/RSJ International Conference on Intelligent Robots and Systems (IROS) (pp. 5792-5799). IEEE.
>
> [5] Tsai, R.Y. and Lenz, R.K., 1989. A new technique for fully autonomous and efficient 3 D robotics hand/eye calibration. IEEE Transactions on robotics and automation, 5(3), pp.345-358.
>
> [6] Furrer, F., Fehr, M., Novkovic, T., Sommer, H., Gilitschenski, I. and Siegwart, R., 2018. Evaluation of combined time-offset estimation and hand-eye calibration on robotic datasets. In Field and Service Robotics (pp. 145-159). Springer, Cham.

---

> > ### Comment · Reviewer_VdxP · 2021-09-02
> > **Thanks for your detailed answers**
> >
> > > We have now included an experiment in our supplementary material to test our best performing method (Direct Regression Model) on three additional grippers in simulation (see section 4.3, lines 155-166, of our supplementary material).
> >
> > Appreciated.
> >
> > > In this sense, a render and compare approach is not fundamentally different from simply a direct regression. Both are end-to-end deep learning methods that regress a rigid transformation.
> >
> > I do not fully agree. A direct regression is a one-shot method predicting a global pose. The render-compare approach only requires estimating relative poses and it can be iteratively applied to refine the pose. It is true that the gripper is not fully visible in the image which might cause some troubles, but the render&compare approach also works very well for object pose estimation with notable occlusions (see "Benchmark for Object Pose Estimation (BOP)" winner CosyPose). The initialization could be some canonical gripper orientation instead of using the direct regression result. It would have been nice to have these results included, but of course in the short rebuttal period it is hard to sufficiently tune the approach.
> >
> > In my opinion, it is not clear whether the classical and DL approaches have been properly tuned and it is questionable whether >1cm stdv is sufficient for many practical applications. However, especially with the added evidence from other (simulated) grippers, the paper is still a good starting point for learning-based hand-eye calibration by testing some standard DL-based approaches. So I would also be okay with an acceptance.

---

> ### Author Response · Authors · 2021-08-30
> **Answer to Reviewer VdxP: Part 3/4**
>
> > The approach of pushing keypoints that are out of the image to the image border is questionable
>
> This is an interesting point. However, we would like to point out that simultaneously predicting which keypoints are visible in an image and their location is still an unsolved research problem.  In early experiments, we attempted to use regressed confidences to determine whether a keypoint is present in an image. These confidences were obtained from our encoder, by finding the maximum of the pre-spatial-soft-argmax features, and passing it through a sigmoid activation, similarly to [1]. However, we have found that this did not perform any better than simply pushing non-present keypoints to the border of an image. As such, we opted to use the latter.
>
>
> > Creating the synthetic training data might be more tedious than doing the external calibration
>
> Creating the synthetic training data does require an initial time overhead. However, once this is done, the approaches we study can readily re-calibrate the camera online from a single image. As such, every time the camera pose changes, due to (1) re-mounting, (2) wear and tear, and (3) collisions between the camera and the environment, the re-calibration is “instant”. In contrast, classical methods doing the external calibration would need to obtain a calibration object, and re-gather data to re-calibrate every time one of the above cases occurs. As such, we believe that in the long run a deep learning-based approach would be more efficient, with the time investment occurring only once.
>
>
> Moreover, we would also like to emphasize that another advantage of our proposed methods over classical external calibration approaches is making the functionality more suitable for non-experts. That is because the dataset collection can be done in the factory/lab a-priori, and in deployment, they do not rely on an external calibration object or data gathering, which makes them more practical for unstructured environments and non-expert users.
>
>
> Finally, We have now included an estimate of how long it took for our dataset to generate (around 30 minutes) and our methods to train (of the order of a few hours, depending on the model) in Section 4 (lines 92-95) of our supplementary material.
>
> > The achieved standard deviation of >1cm does not seem good enough to me
>
> When obtaining results for the classical methods, we experimented with various calibration objects, calibration object pose estimation algorithms, image resolutions, and implementations of the classical baselines. The results presented in our paper correspond to the best performance we managed to achieve with automatic data collection and no human intervention during calibration. Further, our results appear consistent with other works with similar experiments such as [5] or [6] that report RMSE errors > 1cm. Despite this, we acknowledge that the calibration accuracy could have been improved further by manually inspecting the tag detection accuracy on each of the automatically gathered images, only keeping the subset that seems the be the best, or manually annotating the positions of the April Tags by identifying the positions of each corner. This would be extremely tedious and time-consuming and would require a human to be present for each calibration, whilst our investigation focuses on minimising human intervention with automated calibration procedures.
>
> > Figure 4 top doesn't show green/blue lines l325 "sythetic"
>
> We thank the reviewer for the comment and have updated this in the new version of our paper.

---

> ### Author Response · Authors · 2021-08-30
> **Answer to Reviewer VdxP: Part 2/4**
>
> > The paper mentions the Render and Compare approach from Labbe et al. but even though it is SoTA for hand-eye calibration, it is not tried here (I expect this to work better than directly regressing the pose).
>
> Labbé et al. have used a render-and-compare approach to solve the eye-to-hand calibration problem, which is different from our eye-in-hand problem, although it is still related. As their method is not directly applicable to our problem setting, over the past two weeks, we have adapted it to the eye-in-hand calibration problem and used it as a refinement to our Direct Regression Method. After working on this full time for two weeks, we have at best managed to match the results reported in our paper.
>
> We hypothesise that this is because render and compare is not as well suited for eye-in-hand calibration as for eye-to-hand calibration and pose estimation because the object (end-effector) is very close to the camera in the former. First, this causes small changes in the end-effector pose to create very large changes in the appearance of the gripper in the image. Render and compare methods rely on iterative local improvements to the pose in order to be effective, so this effect may hurt their ability to converge. Second, parts of the gripper can be outside of the image plane, making it harder to distinguish between the effects of translation and rotation because it is impossible to observe the entire object's movements. Since detecting a disentangled difference in the rotation, translation and scaling of an object of interest in two input images is a predicate of the success of render and compare methods, we expect them to work worse in this setting.
>
> Finally, we would like to argue about our choice not to use render and compare in the first place. Our paper is to the best of our knowledge the first attempt to apply deep learning to the problem of eye-in-hand calibration. We were interested in studying how fundamentally different ways to use deep learning to tackle this issue would respond to the task, and what the potential pitfalls of each may be. In this sense, a render and compare approach is not fundamentally different from simply a direct regression. Both are end-to-end deep learning methods that regress a rigid transformation. As such, we instead chose to compare a standard end-to-end approach (direct regression model), with a sparse 2D feature extraction and 2D-3D correspondence based approach (sparse correspondence model), and a dense image prediction with 3D registration-based approach (dense correspondence model).
>
>
> > The assumption that the gripper is always in view is quite strong. In this case we could also just attach a marker to the gripper.
>
>
> We believe that for an eye-in-hand setup, it is not uncommon to have the gripper in view to allow for precise manipulation, as in [1], [2], [3] and [4].
>
> Moreover, getting part of the gripper in view when mounting a camera to the wrist naturally occurs for most mounting angles and reasonable camera fields of view. Forcing the gripper to be out of frame would probably result in less functional camera angles. We also believe that attaching a marker to the gripper would not always be practical. Most grippers have very small flat surfaces on which to attach a marker, if any. For these, the viewing angle would also typically be quite extreme in our setting. Given this, it is not obvious that a precise marker pose can be obtained in practice from simply attaching it to a gripper. Moreover, even if it were possible to do so, this would return the marker pose in the camera frame. To get the camera calibration,  the marker pose relative to the end-effector would still need to be obtained, and it is not obvious how this could be achieved.
>
> > For the constrained and un-occluded pose space of the gripper I would also expect classical edge matching techniques to work well
>
> We acknowledge that it may be possible to hand engineer other solutions for our setting. In early experiments, we attempted a similar idea that relied on an affordance segmentation matching algorithm. Precisely, we used the segmentation masks of the gripper and of the gripper’s model rendered at a pose proposal to obtain a cost function based on the Intersection over Union of the two. We then used this cost function and the cross entropy method in order to optimise the gripper pose. Overall, we found that it did not produce any convincing results, as we found the optimisation could find solutions that would decrease the cost we defined but not necessarily get to the correct extrinsics calibration at the end. As such, we ended up dropping this as a potential avenue for further investigation.

---

> ### Author Response · Authors · 2021-08-30
> **Answer to Reviewer VdxP: Part 1/4**
>
> We thank the reviewer for their thorough feedback and constructive criticism. We have uploaded a new version of our paper and supplementary material, with changes highlighted in blue. Please find our answers below to the specific concerns raised.
>
> > The resolution of the RGB sensor is not mentioned even though it has quite some impact on calibration results
>
> The resolution of the RGB sensor was 480x848. We have used this resolution for the classical methods and downsampled it to 144x256 for the learning-based methods. We have now updated our paper with this information in lines 199-201.
>
> > A single April Tag is not really adequate to get a precise calibration. In my experience calibration boards, e.g. the Charuco board lead to higher accuracies than reported in the paper with classical methods
>
> During this project, we spent significant time and effort exploring three different calibration objects: April Tags, ArUco Tags and ChArUco boards. We have also experimented with various implementations of calibration object pose estimation algorithms. Overall, we have found that the AprilTag3 library used to estimate the pose of a large April Tag performed best.
>
>
> > There is only one robotic experiment using a single camera-robot combination. More evidence is needed that this method consistently works well.
>
> We have now included an experiment in our supplementary material to test our best performing method (Direct Regression Model) on three additional grippers in simulation (see section 4.3, lines 155-166, of our supplementary material). These new grippers consist of a two-finger RG2 gripper, a three-finger Jaco Hand gripper and a Baxter suction gripper. In all three, our results are consistent with the results reported in our paper, which indicates there is nothing particular about the gripper tested in the paper and that the method would work consistently well on different hardware.

---

### Official Review · Reviewer_mbiM · 2021-07-27

**Originality:** Good
**Technical Quality:** Very Good
**Clarity Of Presentation:** Good
**Impact:** 3

**Recommendation:**

Strong Accept: I recommend accepting the paper and will argue for my recommendation even if other reviewers hold a different opinion.

**Summary:**

The present work considers the eye-in-hand camera calibration problem, i.e. the problem of estimating the relative pose between a camera's coordinate frame mounted on an end-effector and the end-effector's coordinate frame. The authors evaluate three different learning-based models and compare them against numerous classical baselines. All training is done on synthetic data and evaluated on a real-world platform. The work reports on the surprising finding that direct regression exhibits the best performance.

**Issues:**

__General__

- It is claimed (in. l.42) that  _"this calibration can be estimated directly from the camera’s image itself, without requiring any external apparatus, as long as the image can observe part of the end-effector, such as the gripper’s fingers."_  Do I understand correctly that this ends up memorizing a specific end-effector and would not generalize to using a different end-effector without retraining?
- The fact that geometric approaches perform worse is surprising to me as well. I wonder whether this is architecture related and experimenting with different types of architectures would have resulted in a better performance. The work mentions the sensitivity to outliers as a potential reason. Maybe using alignment methods that are robust to outliers (such as the methods developed Luca Carlone's group) could help?
- How were the classical geometric methods integrated into the learning pipeline? Did you use differentiable variants of these methods?

__Appendix__

- It might be easier understandable to present the network architecture parameters in a table as, e.g., on p.2 of [this supplementary](https://www.robots.ox.ac.uk/~namhoon/doc/DESIRE-supp.pdf).

__Code__

- The instructions in the code repository seem to be (at least slightly) incorrect or incomplete. E.g. they say to execute `python direct_regression_model.py` but the file `direct_regression_model.py` does not exist. I assume you meant `example_direct_regression_model.py`?
- Consider using [this codebase template](https://github.com/paperswithcode/releasing-research-code) for code publishing.

**Reviewer Expertise:**

Fair: Some knowledge of the area

**Strengths And Weaknesses:**

__Strengths__

- _Motivation:_ The problem considered in this work is highly relevant and well-motivated. It may be worth discussing why non deep-learning methods should fail here.
- _Results_: The results compared to state-of-the-art seem convincing.
- _Video:_ While the video would benefit from some narration and some more details, it still gives a nice first (teaser-style) overview.

__Weaknesses__

- _Literature:_ Numerous general and several specific related works are missing (e.g.  _"Fast Eye-in-Hand 3-D Scanner-Robot Calibration for Low Stitching Errors"_). Some of the works consider even non-overlapping views, e.g. for a depth camera (e.g.  _"Recursive Bayesian Calibration of Depth Sensors with Non-Overlapping Views"_). While I understand, that it is impossible to discuss all of the related work in the calibration space, CoRL does not impose limitations on reference pages and thus, the related work discussion could be much more extensive at least with regards to closely related works.
- _Ablation:_ The work is missing several potential ablations. For example, the use of different orientation representations might impact performance. Similarly, it is not clear whether hyperparameter tuning was performed and to what extent. The work seems to suggest that this was not the case.
- _Baselines:_ Why using OpenCV's baselines is meaningful, I am not sure if they provide the most recent results and I wonder if newer works already outperform these baselines.

**Summary Of Recommendation:**

Overall, I believe this work to be interesting and worthy of publication. The method is useful for numerous real-world problems and opens up a relevant research direction of using deep-learning in eye-in-hand calibration. The work could benefit from providing more ablations to better understand the surprising phenomena of successful direct regression but all in all I tend to support its publication.

---

> ### Author Response · Authors · 2021-08-30
> **Answer to Reviewer mbiM : Part 4/4**
>
> References
>
> [1] Zhou, Y., Barnes, C., Lu, J., Yang, J. and Li, H., 2019. On the continuity of rotation representations in neural networks. In Proceedings of the IEEE/CVF Conference on Computer Vision and Pattern Recognition (pp. 5745-5753).
>
> [2] Daniilidis, K., 1999. Hand-eye calibration using dual quaternions. The International Journal of Robotics Research, 18(3), pp.286-298.
>
> [3] Tsai, R.Y. and Lenz, R.K., 1989. A new technique for fully autonomous and efficient 3 D robotics hand/eye calibration. IEEE Transactions on robotics and automation, 5(3), pp.345-358.
>
> [4] Horaud, R. and Dornaika, F., 1995. Hand-eye calibration. The international journal of robotics research, 14(3), pp.195-210.
>
> [5] Park, F.C. and Martin, B.J., 1994. Robot sensor calibration: solving AX= XB on the Euclidean group. IEEE Transactions on Robotics and Automation, 10(5), pp.717-721.
>
> [6] Furrer, F., Fehr, M., Novkovic, T., Sommer, H., Gilitschenski, I. and Siegwart, R., 2018. Evaluation of combined time-offset estimation and hand-eye calibration on robotic datasets. In Field and Service Robotics (pp. 145-159). Springer, Cham. Available at: https://github.com/ethz-asl/hand_eye_calibration [Accessed 30 Aug. 2021].
>
> [7] GitHub. 2021. GitHub - ros-planning/moveit_tutorials: A sphinx-based centralized documentation repo for MoveIt. [online] Available at: <https://github.com/ros-planning/moveit_tutorials> [Accessed 30 August 2021].
>
> [8] GitHub. 2021. GitHub - IFL-CAMP/easy_handeye: Automated, hardware-independent Hand-Eye Calibration. [online] Available at: <https://github.com/IFL-CAMP/easy_handeye> [Accessed 30 August 2021].
>
> [9] GitHub. 2021. GitHub - jhu-lcsr/handeye_calib_camodocal: Easy to use and accurate hand eye calibration which has been working reliably for years (2016-present) with kinect, kinectv2, rgbd cameras, optical trackers, and several robots including the ur5 and kuka iiwa.. [online] Available at: <https://github.com/jhu-lcsr/handeye_calib_camodocal> [Accessed 30 August 2021].
>
> [10] GitHub. 2021. GitHub - crigroup/handeye: ROS package for calibrating sensors to a known reference frame.. [online] Available at: <https://github.com/crigroup/handeye> [Accessed 30 August 2021].
>
> ‌
> ‌

---

> > ### Comment · Reviewer_mbiM · 2021-09-03
> > **Your Answers**
> >
> > Thank you for answering all my questions in detail. I still believe that the work is worthy of publication and will slightly increase my recommendation as several of my concerns have been addressed.

---

> ### Author Response · Authors · 2021-08-30
> **Answer to Reviewer mbiM : Part 3/4**
>
> > The work mentions the sensitivity to outliers as a potential reason. Maybe using alignment methods that are robust to outliers (such as the methods developed Luca Carlone's group) could help?
>
> We believe that such methods would not necessarily help further, since the poor performance of the keypoint-based method is not due to a large amount of noise on a minority of outlier keypoints, but instead on how sensitive the method is to a small amount of noise on the entirety of the keypoints. In our controlled noise experiments, we found that RANSAC was able to root out outlier keypoints that have a large amount of noise. However, when small amounts of noise was added to all keypoints, pose estimation performance with PnP deteriorated very quickly. This is shown in the Analysis section of our paper.
>
> > How were the classical geometric methods integrated into the learning pipeline? Did you use differentiable variants of these methods?
>
> We have used a non-differentiable variant of ICP and RANSAC + PnP. For this reason, these methods were not part of the learning pipelines. The learned methods were trained to obtain the inputs required by the geometric methods. Integrating differentiable variants of these methods is an interesting avenue for future work that would enable the learned methods to be trained end-to-end. In this work, we focussed on exploring how the most typical approaches one would consider when using deep learning on this task perform and what pitfalls they would have.
>
>
> > It might be easier understandable to present the network architecture parameters in a table as, e.g., on p.2 of this supplementary.
>
> We appreciate the suggestion and have incorporated this change in our supplementary material.
>
> > The instructions in the code repository seem to be (at least slightly) incorrect or incomplete. E.g. they say to execute python direct_regression_model.py but the file direct_regression_model.py does not exist. I assume you meant example_direct_regression_model.py? Consider using this codebase template for code publishing.
>
> We thank the reviewer for pointing this out. We have now corrected our submitted code.

---

> ### Author Response · Authors · 2021-08-30
> **Answer to Reviewer mbiM : Part 2/4**
>
> > Baselines: Why using OpenCV's baselines is meaningful, I am not sure if they provide the most recent results and I wonder if newer works already outperform these baselines.
>
> OpenCV implemented methods that correspond to the standard references in terms of eye-in-hand calibration (Dainiilidis [2], Tsai and Lenz [3], Harod and Doraika [4], Park and Martin [5]). To the best of our knowledge, all public eye-in-hand calibration implementations for RGB cameras at the very least use one of these methods as their backbone. For instance, [6], [7], [8],  [9] and [10] implement or rely on one or more of these methods. We could not find any public implementations of more recent methods, but we also believe the ones we tested are the most meaningful since they are the most widely used. Finally, OpenCV was simply chosen as a reliable implementation of these baselines.
>
>
> > It is claimed (in. l.42) that "this calibration can be estimated directly from the camera’s image itself, without requiring any external apparatus, as long as the image can observe part of the end-effector, such as the gripper’s fingers." Do I understand correctly that this ends up memorizing a specific end-effector and would not generalize to using a different end-effector without retraining?
>
> Yes, the presented methods are considering a single end-effector, and changing the end-effector would require re-training. However, for that single end-effector, these can be used to automatically calibrate a camera online from a single image without needing an external calibration object and time-consuming data collection. Moreover, it is quite typical for robot learning methods to be trained on a single robot, and the method can easily be re-trained on each new hardware. In our case, this can be done in a couple of hours, simply by having a model of the gripper. The question of generalising robot learning approaches to different robots (in our case, grippers) without re-training is an open research problem, and may be an interesting avenue for future work.
>
>
> > The fact that geometric approaches perform worse is surprising to me as well. I wonder whether this is architecture related and experimenting with different types of architectures would have resulted in a better performance.
>
> We also considered this, and as such, when designing our architectures, we experimented with several alternatives for both geometric models. This includes residual layers, parametric and non-parametric upsampling, different activation functions, regularisation methods, network depth, and input resolutions. For the sparse correspondence model, we have also experimented with various ways of dealing with the fact that not all keypoints are present in all images. The architectures we present in our paper correspond to the best designs from all those considered, and we believe that the reasons these methods performed worse than the direct regression model are the fundamental and theoretical issues we identified in the analysis section of our paper, not the specific network architectures used.

---

> ### Author Response · Authors · 2021-08-30
> **Answer to Reviewer mbiM : Part 1/4**
>
> We thank the reviewer for their time and insightful comments and suggestions. We have now uploaded a new version of the paper with changes appearing in blue. Below are specific answers to the points raised.
>
> > Motivation: The problem considered in this work is highly relevant and well-motivated. It may be worth discussing why non deep-learning methods should fail here.
>
> We thank the reviewer for the suggestion. Non-deep learning methods should fail since they require a dedicated external calibration object and re-calibrating every time the camera's position changes. This makes them unsuitable for non-experts as well as impractical, since re-calibration needs to be performed every time the camera position changes due to (1) re-mounting, (2) wear and tear, and (3) collisions between the camera and the environment. Deep learning methods on the other hand have the potential to be trained only once, and to re-calibrate a camera online from a single image every time its position changes for any of the aforementioned reasons. We have now updated the Introduction of our paper in lines 35-41 to reflect this change.
>
> > Literature: Numerous general and several specific related works are missing (e.g. "Fast Eye-in-Hand 3-D Scanner-Robot Calibration for Low Stitching Errors"). Some of the works consider even non-overlapping views, e.g. for a depth camera (e.g. "Recursive Bayesian Calibration of Depth Sensors with Non-Overlapping Views"). While I understand, that it is impossible to discuss all of the related work in the calibration space, CoRL does not impose limitations on reference pages and thus, the related work discussion could be much more extensive at least with regards to closely related works.
>
> We appreciate the suggestion and have updated the related work section (lines 80-85) with several other related papers, including the recommended.
>
> > Ablation: The work is missing several potential ablations. For example, the use of different orientation representations might impact performance.
>
> We thank the reviewer for the constructive criticism of our work. Whilst these may be interesting experiments, we believe they would distract from the main focus of our paper,  which is to investigate how structurally different deep-learning-based approaches (end-to-end, deep-learning-enabled sparse and dense correspondences) would perform in this task as well as identifying any potential pitfalls they may have. Regarding the orientation representation, we have followed the suggestions presented in [1], which we believe presents a convincing account of how neural networks respond to different orientation representations.
>
> > Similarly, it is not clear whether hyperparameter tuning was performed and to what extent. The work seems to suggest that this was not the case.
>
> We would like to clarify our point in the paper about the tuning of our different proposed methods. We have spent a significant amount of time tuning all our methods to the best of our ability, including architecture and training hyperparameter searches,  while attempting to spend roughly equal amounts of time and resources tuning them. If larger and larger searches were to be carried out, it may be possible that the performance of any method could increase. However, we do not expect this to alter their relative performance and the results and conclusions we presented in the paper in any significant way. Instead, we believe we have identified the most important limitations for each method in our task setting, which we describe in the Analysis section of our paper.

---

### Meta-Review · Area_Chair_yV4L · 2021-08-11

**Recommendation:** Accept (Poster)
**Confidence:** 5

**Metareview:**

The proposed direct regression method for eye-in-hand calibration is clearly interesting, particularly because it seems to give very good results and even outperforms classical geometrical methods. All reviewers agree in that this is an approach that is worth considering, but they also state that the comparison to the correspondence-based methods could be more elaborated. As it is, this comparison seems a bit unfair, both in terms of exploiting more the potential of the competing methods (e.g. hyperparameter tuning), and also regarding the actual overall effort required to get the aimed results. For the latter, it clearly needs to be taken into account that training data set generation is also an important step, which seems to be necessary here everytime the setup changes. It is definitely nice to see that the purely learning-based approach outperforms the more classical methods, but it is hard to draw a general conclusion out of this from the given experimental evaluations. The fact that only one specific gripper is used in the simulation and that it is required to always see the gripper in the images also limits this attempt of drawing a general conclusion. In their answer, the authors might consider whether a somewhat weaker claim still highlights the benefit of their proposed approach, but also mention in more detail relations to the recent Render & Compare method by Labbé et al.

Post-rebuttal:
In their answers to the reviewers, the authors could clarify most of the raised questions. In the updated version of the paper, the work clearly represents a valuable contribution, meeting the criteria for CoRL.

---

> ### Author Response · Authors · 2021-08-30
> **Answer to Area Chair yV4L : Part 2/2**
>
> > [...] and also regarding the actual overall effort required to get the aimed results. For the latter, it clearly needs to be taken into account that training data set generation is also an important step,[...]
>
> Regarding the effort required to get the aimed results, we have now clarified this point in the introduction section of our paper (lines 39-40) and section 4 of our supplementary material (lines 92-96).
>
> Specifically, in the introduction, we specify that the deep learning methods we present do require an initial overhead to generate training data and train neural network models. Nonetheless, once a model is trained for a particular gripper, it can be used to readily calibrate the camera.  As such, every time the camera pose changes, due to (1) re-mounting, (2) wear and tear, and (3) collisions between the camera and the environment, the re-calibration is “instant”. In contrast, classical methods doing the external calibration would need to obtain a calibration object, and re-gather data to re-calibrate every time one of the above cases occurs.
>
> We would also like to emphasize that another advantage of our proposed methods over classical calibration approaches is that it would make eye-in-hand calibration more accessible to non-experts. That is because the dataset collection can be done in the factory/lab a-priori, and in deployment, they do not rely on an external calibration object or data gathering, which makes them more practical for unstructured environments and non-expert users.
>
> Finally, in our supplementary material, we have now included an estimate of how long it took for our dataset to generate (around 30 minutes since this is a highly parallelizable process) and our methods to train (around 30 minutes for the direct regression model, 3.5 hours for the sparse correspondence model and 3 hours for the dense correspondence model.)
>
> > [...]  which seems to be necessary here everytime the setup changes.
>
> As we specified above, this would be necessary every time the hardware changes. However, then it can be readily used for a particular hardware when the camera-to-gripper pose changes.
> > In their answer, the authors might consider whether a somewhat weaker claim still highlights the benefit of their proposed approach, [...]
>
> We appreciate the suggestion and have now updated the instruction section of our paper accordingly.
>
> Specifically, in lines 34-42, we better highlight the benefits of applying deep learning to eye-in-hand camera calibration over using classical alternatives. This includes (1) not relying on an external calibration object, which is more practical and can enable the calibration task to be performed by non-experts, and (2) obtaining the ability to readily re-calibrate the camera online every time the camera pose changes (due to re-mounting, wear and tear or collisions), which is more practical than classical methods which need to collect a new calibration dataset every time.
>
> Finally, in lines 59-62 of our paper, we updated our contributions to convey that (1) our investigation shows that deep learning approaches constitute a viable option for eye-in-hand camera calibration, and as such open up this field as an interesting avenue of investigation, (2) end-to-end approaches have the potential to outperform classical alternatives, and (3) we study different ways deep learning could be applied to the problem, and identify key weaknesses of more geometrically constrained approaches.
>
> > [...] but also mention in more detail relations to the recent Render & Compare method by Labbé et al.
>
> Labbé et al. tackle the task of simultaneously learning a robot's pose relative to an external camera (eye-to-hand camera calibration) and the robot's joint configuration from a single image. Their method is not directly applicable to our setting. Moreover, it is not fundamentally different from attempting a direct regression of the camera pose as both methods rely on end-to-end learning. In contrast, the (sparse and dense) correspondence models we study rely on well-established geometric methods and we use deep learning to provide key missing components. As such, we opted to study and understand how end-to-end learning does versus deep learning enhanced geometric methods and have identified their strengths and weaknesses. Nonetheless, in the past two weeks, we have attempted to adapt a render a compare type approach to our direct regression method as a refinement step. However, we have not managed to improve the performance of simple direct regression. We believe this may have to do with the particular constraints of our problem setting, which differ from the setting studied by Labbé and the typical pose estimation setup render and compare approaches were developed on.

---

> ### Author Response · Authors · 2021-08-30
> **Answer to Area Chair yV4L : Part 1/2**
>
> We thank the area chair for their time and their feedback. We have updated our paper and supplementary material with changes highlighted in blue, incorporating the feedback provided by each reviewer.
>
> Please find a summary of our contributions below:
> - We introduce deep learning to the problem of eye-in-hand calibration.
> - We identify, implement and evaluate three natural ways deep learning can enable online eye-in-hand calibration from a single image. This includes an end-to-end approach and two well established geometric approaches where deep learning is used to provide key missing inputs.
> - We benchmark all proposed methods against well established and widely used classical methods both in simulation and the real world.
> - We identify and analyse the most important pitfalls that arise when using the deep learning-enabled geometric methods that we studied, which open up interesting avenues for future work.
> - We show the potential of end-to-end deep learning for online eye-in-hand camera calibration in everyday environments. This includes the possibility to calibrate the camera without a calibration object or test-time dataset collection whilst outperforming typical classical approaches.
> - We have now tested the applicability of our best performing method to other hardware in simulation, as requested by one of the reviewers.
>
> Here are specific answers to the points raised:
>
> > All reviewers agree in that this is an approach that is worth considering, but they also state that the comparison to the correspondence-based methods could be more elaborated. As it is, this comparison seems a bit unfair, both in terms of exploiting more the potential of the competing methods (e.g. hyperparameter tuning), [...]
>
> As we mentioned in the individual reviewer responses,  we would like to clarify our point in the paper about the tuning of our different proposed methods. We have spent a significant amount of time tuning all our methods to the best of our ability, including architecture and training hyperparameter searches,  while attempting to spend roughly equal amounts of time and resources tuning them. If larger and larger searches were to be carried out, it may be possible that the performance of any method could increase. However, we do not expect this to alter their relative performance and the results and conclusions we presented in the paper in any significant way. Instead, we believe we have identified the most important limitations for each method in our task setting, which we describe in the Analysis section of our paper.

---

### Decision · Program_Chairs · 2021-09-13

**Decision:**

Accept (Poster)

**Comment:**

The proposed direct regression method for eye-in-hand calibration is clearly interesting, particularly because it seems to give very good results and even outperforms classical geometrical methods. All reviewers agree in that this is an approach that is worth considering, but they also state that the comparison to the correspondence-based methods could be more elaborated. As it is, this comparison seems a bit unfair, both in terms of exploiting more the potential of the competing methods (e.g. hyperparameter tuning), and also regarding the actual overall effort required to get the aimed results. For the latter, it clearly needs to be taken into account that training data set generation is also an important step, which seems to be necessary here everytime the setup changes. It is definitely nice to see that the purely learning-based approach outperforms the more classical methods, but it is hard to draw a general conclusion out of this from the given experimental evaluations. The fact that only one specific gripper is used in the simulation and that it is required to always see the gripper in the images also limits this attempt of drawing a general conclusion. In their answer, the authors might consider whether a somewhat weaker claim still highlights the benefit of their proposed approach, but also mention in more detail relations to the recent Render & Compare method by Labbé et al.

Post-rebuttal:
In their answers to the reviewers, the authors could clarify most of the raised questions. In the updated version of the paper, the work clearly represents a valuable contribution, meeting the criteria for CoRL.